# Structural basis of substrate recognition by a polypeptide processing and secretion transporter

**Virapat Kieuvongngam[1], Paul Dominic B Olinares[2], Anthony Palillo[1†], Michael L Oldham[1,3], Brian T Chait[2], Jue Chen[1,3]***

[1]Laboratory of Membrane Biophysics and Biology, The Rockefeller University, New York, United States; [2]Laboratory of Mass Spectrometry and Gaseous Ion Chemistry, The Rockefeller University, New York, United States; [3]Howard Hughes Medical Institute, Chevy Chase, United States

**Abstract** The peptidase-containing ATP-binding cassette transporters (PCATs) are unique members of the ABC transporter family that proteolytically process and export peptides and proteins. Each PCAT contains two peptidase domains that cleave off the secretion signal, two transmembrane domains forming a translocation pathway, and two nucleotide-binding domains that hydrolyze ATP. Previously the crystal structures of a PCAT from *Clostridium thermocellum* (PCAT1) were determined in the absence and presence of ATP, revealing how ATP binding regulates the protease activity and access to the translocation pathway. However, how the substrate CtA, a 90-residue polypeptide, is recognized by PCAT1 remained elusive. To address this question, we determined the structure of the PCAT1-CtA complex by electron cryo-microscopy (cryo-EM) to 3.4 Å resolution. The structure shows that two CtAs are bound via their N-terminal leader peptides, but only one is positioned for cleavage and translocation. Based on these results, we propose a model of how substrate cleavage, ATP hydrolysis, and substrate translocation are coordinated in a transport cycle.

**\*For correspondence:**
juechen@rockefeller.edu

**Present address:** [†]The Weill Cornell Graduate School of Medical Sciences, Weill Cornell Medicine, New York, United States

**Competing interests:** The authors declare that no competing interests exist.

## Introduction

ATP-binding cassette (ABC) transporters are ubiquitous membrane proteins that use the chemical energy of ATP to transport substrates across biological membranes (*Locher, 2016*; *Srikant and Gaudet, 2019*). Substrates of ABC transporters range from small ions to large lipids and proteins. The peptidase-containing ABC transporters (PCATs) are unique as they contain both a peptidase domain and a transporter core. In gram-positive bacteria, PCATs function both as maturation proteases and exporters to secrete quorum-sensing or antimicrobial polypeptides (*Fath and Kolter, 1993*; *Håvarstein et al., 1995*). In gram-negative bacteria, PCATs interact with two other membrane proteins to form the Type I secretion system (T1SS), a continuous channel penetrating both the inner and outer membranes (*Binet et al., 1997*; *Michiels et al., 2001*; *Thomas et al., 2014*).

Although PCATs were first described 20 years ago and are essential to prokaryotic life (*Håvarstein et al., 1995*), structural and functional studies of these proteins have been limited. PCATs are typically homodimers; each subunit consists of a transmembrane domain (TMD), a nucleotide-binding domain (NBD) characteristic of all ABC transporters, and a unique C39 cysteine peptidase (PEP) domain essential in recruiting the substrate. In Gram-positive bacteria, substrates of PCATs are synthesized as precursors with an N-terminal leader peptide and a C-terminal cargo peptide. Proteolytic cleavage of the leader peptide at the conserved double-glycine motif is necessary for cargo secretion (*Nishie et al., 2011*). The substrates of the T1SS in Gram-negative bacteria are

typically large proteins containing a C-terminal secretion signal that is not subjected to proteolytic processing (*Lecher et al., 2012*).

What are the structural features that enable PCATs to conduct protein substrates? Are they typical ATP-driven pumps like most ABC transporters or do they form an ATP-gated channel akin to CFTR? Crystal structures of isolated peptidase domains and NBDs have been reported for several PCATs (*Lecher et al., 2012*; *Ishii et al., 2010*; *Ishii et al., 2013*; *Schmitt et al., 2003*; *Bobeica et al., 2019*). Recently, the structures of a full-length transporter, PCAT1 from *Clostridium thermocellum,* were determined by X-ray crystallography (*Lin et al., 2015*). The structure of PCAT1 in the absence of ATP and substrate reveals a large α-helical barrel sufficient to accommodate a small protein. Typical of an inward-facing ATP transporter, the protein-secretion pathway is open to the cytosol and closed to the extracellular side. The NBDs are semi-separated and the PEP domains dock onto the intracellular openings of the translocation pathway. The structure of an ATP-bound PCAT1 shows a closed NBD dimer and an occluded translocation pathway. The two PEP domains are not resolved in the structure, suggesting that they are flexibly attached to the transporter core. A key feature that renders CFTR an ion channel instead of a transporter is the 'broken' intracellular gate: in the NBD-dimerized conformation, an opening in the TM helical bundle connects the ion-conduction pathway to the cytosol (*Zhang et al., 2017*). This feature is not observed in PCAT1. Like other ATP-driven pumps (*Dawson and Locher, 2006*; *Kim and Chen, 2018*; *Johnson and Chen, 2018*; *Hofmann et al., 2019*), the intracellular gate of PCAT1 is closed off upon NBD-dimerization. Thus, we suggest that PCAT1 functions through the classic alternating-access mechanism.

The PCAT1 substrate, CtA, is a small protein consisting of a 24-residue leader peptide followed by a 66-residue cargo peptide. The protease activity of PCAT1 is specific and is inhibited by ATP-binding (*Lin et al., 2015*). Different from most ABC transporters, the presence of substrate does not stimulate ATP hydrolysis of PCAT1 (*Lin et al., 2015*). To understand how PCAT1 interacts with its substrate, we characterized the PCAT1-CtA complex by native mass spectrometry (MS) and electron cryo-microscopy (cryo-EM). The structure of the PCAT1-CtA complex reveals asymmetric positioning of two substrates, supporting a model for strict coupling of cleavage and translocation.

**Table 1.** Mass measurements from native MS analysis of PCAT1 samples.

| Sample | Expected mass (Da) | *In UDM* | | | *In C$_8$E$_4$* | | |
| | | Measured Mass ± SD[a] (Da) | Mass error (%)[b] | | Measured Mass ± SD[a] (Da) | Mass error (%)[b] | |
| --- | --- | --- | --- | --- | --- | --- | --- |
| wt PCAT1 dimer | 162,176 | 162,175 ± 1 | 0.0008 | - | 162,171 ± 1 | 0.003 | - |
| CtA | 10,207.9 | 10,206.5 ± 0.5 | 0.014 | | 10,206.5 ± 0.3 | 0.014 | |
| - | - | - | - | % Abundance[c] | - | - | % Abundance[c] |
| C21A-PCAT1 dimer + 1 CtA | 172,320 | 172,337 ± 4 | 0.010 | 25 ± 2 | 172,341 ± 3 | 0.012 | 16 ± 2 |
| C21A-PCAT1 dimer + 2 CtA | 182,528 | 182,549 ± 3 | 0.012 | 75 ± 2 | 182,558 ± 6 | 0.017 | 84 ± 2 |

[a] From the average and S.D. of the mass values calculated from each charge state peak within a charge-state distribution (n ≥ 4 charge states).

[b] Mass accuracy measured by the relative difference between measured and expected masses divided by the expected mass.

[c] Based on relative peak intensities for the peak series corresponding to the 182 kDa and 172 kDa complexes within the same MS spectrum and corrected for gas-phase dissociation (see *Figure 1—figure supplement 2* for more details) (n ≥ 3 conditions).

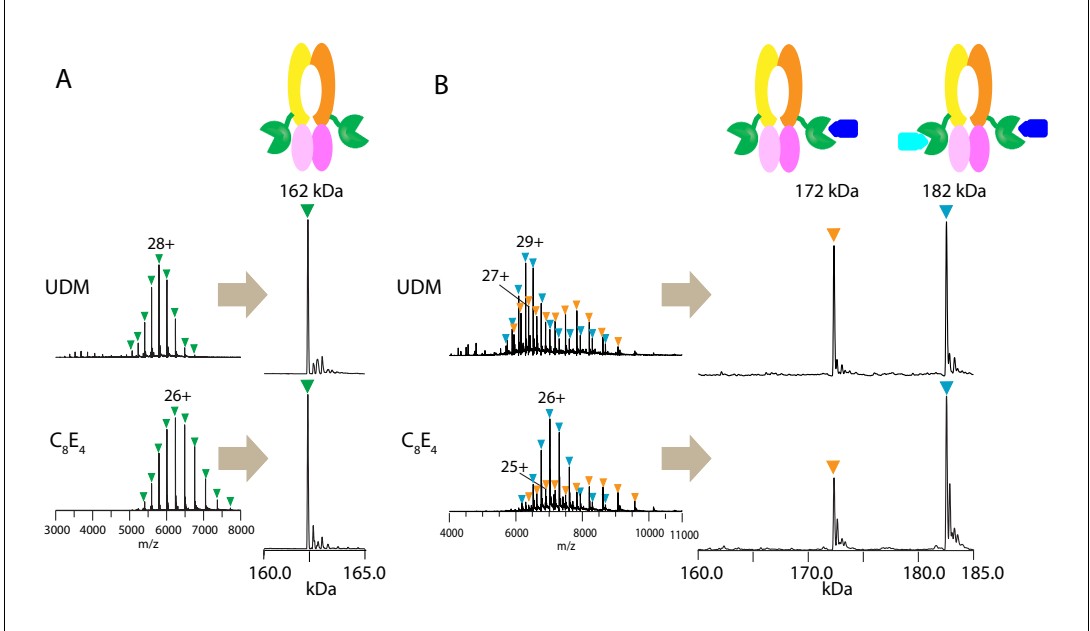

**Figure 1.** Native mass spectrometry analysis of wt PCAT1 and PCAT1(C21A)-CtA complexes. (**A**) *wt* PCAT1 and (**B**) PCAT1(C21A)-CtA complex. The samples were analyzed in 200 mM ammonium acetate containing either UDM or $C_8E_4$ at 2X critical micelle concentration (CMC). The peak series for the protein complex were at lower charge states in $C_8E_4$ (charged-reduced) than in UDM, consistent with previous native MS results of other membrane protein complexes (*Reading et al., 2015*). Note that the peak intensities for the 172 kDa complex in the deconvoluted spectra in (**B**) includes both the complex present in-solution and the subcomplex resulting from the gas-phase dissociation of the 182 kDa complex upon collision activation (for estimation of the % relative abundances of the complexes with correction for gas-phase dissociation, see *Figure 1—figure supplement 2*). The online version of this article includes the following figure supplement(s) for figure 1:

**Figure supplement 1.** Native MS analysis of the *wt* PCAT1 at varying activation energies.
**Figure supplement 2.** Native MS analysis of the PCAT1(C21A) transporter in complex with CtA at varying activation energies.

## Results

### Stoichiometry determination using native mass spectrometry

PCAT1 contains two identical PEP domains, raising the question of whether two copies of the substrate can bind and be translocated simultaneously. To address this question, we used native MS to determine the maximum binding capacity of PCAT1.

First, we tested if the PCAT1 homodimer remains intact during native MS analysis in two different detergents: *n*-undecyl-β-D-maltopyranoside (UDM) and octyl tetraethylene glycol ether ($C_8E_4$). Previous structural and biochemical characterization of the PCAT1 transporter was performed in UDM (*Lin et al., 2015*). $C_8E_4$ has been shown to require the lowest activation energy for detergent removal in the gas phase with minimal destabilization of membrane protein complexes during MS analysis (*Reading et al., 2015*). The mass spectra obtained in both detergents at optimal MS parameters (*Figure 1—figure supplement 1*) yielded a single peak series with a measured mass consistent with the mass of dimeric PCAT1 (*Figure 1A* and *Table 1*).

To obtain a stable PCAT1-CtA complex, we used a proteolytic-deficient mutant (C21A) that can bind but does not cleave the substrate (*Lin et al., 2015*). Two-fold molar excess of the substrate was mixed with PCAT1 prior to MS analysis. In both detergents, we observed two main assemblies of 172.3 and 182.5 kDa, corresponding to dimeric PCAT1 bound to one or two CtA, respectively (*Figure 1B*, *Table 1*). Based on the relative peak intensities with correction for gas-phase dissociation, the PCAT1 dimer with two CtA bound (1:2 complex) is the dominant assembly in the presence of excess CtA (75 ± 2% in UDM and 84 ± 2% in $C_8E_4$) (*Figure 1—figure supplement 2* and *Table 1*). The lower abundance of the 1:2 complex in UDM is likely due to the higher collision activation energies needed to remove UDM that lead to increased protein complex dissociation (*Figure 1—figure supplements 1* and *2*).

## Cryo-EM structure of the PCAT1-substrate complex

Next, we determined the cryo-EM structure of the PCAT1(C21A)-CtA complex in the presence of excess CtA and in the absence of ATP. The overall resolution is approximately 3.4 Å (*Figure 2*, *Table 2*, *Figure 2—figure supplements 1–4*). Side-chain densities are prominent in the TMDs, allowing unambiguous residue assignment (*Figure 2—figure supplement 3*). The density of the PEPs and NBDs show clear definition of secondary structure, permitting docking of the crystal structures of isolated PEP and NBD. Extra densities, unaccounted by PCAT1, are found on the surface of both PEP domains, likely corresponding to the N-terminal region of CtA (*Figure 2—figure supplement 3*). The final model, containing residues 8–722 of PCAT1 and two leader peptides of CtA, was refined against the EM density to excellent geometry and statistics (*Table 2*).

Consistent with the native MS analysis, two copies of CtA were bound to a single PCAT1 transporter (*Figure 2*). In this ATP-free, substrate-bound conformation, the TM cavity is continuous with the cytosol and closed off to the extracellular space. The two PEP domains, each binding a CtA molecule, interact with the transporter core at the TMD/NBD interface. The structure is two-fold symmetric, with the exception of the C-terminal regions of the substrate, a point that we will discuss in detail below.

## A conserved catalytic mechanism

The sequence of the CtA leader peptide is homologous to that of other PCAT substrates in Gram-positive bacteria (*Figure 3A*). The cleavage site contains a consensus sequence of L(−12)XXXE(−8)L

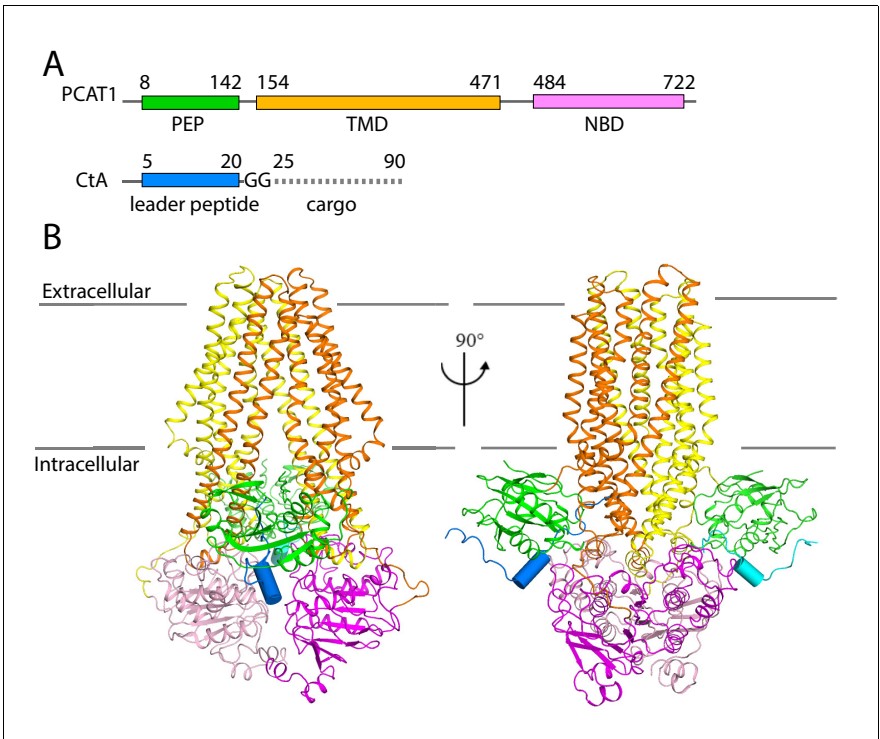

**Figure 2.** The cryo-EM structure of PCAT1-CtA complex. (**A**) Cartoon illustration depicting domain organization of PCAT1 and its substrate, CtA. The symbol GG denotes the double glycine motif. The dotted line represents the unstructured cargo region. (**B**) Two orthogonal views of the PCAT1-CtA complex structure, the leader peptide helix of CtA is shown as a cylinder. The cartoon is color-coded by domains. Blue, CtA; magenta, NBD; yellow, TMD; green, PEP.

The online version of this article includes the following figure supplement(s) for figure 2:

**Figure supplement 1.** Summary of image processing workflow.
**Figure supplement 2.** Local resolution estimation of the cryo-EM density map.
**Figure supplement 3.** The cryo-EM density of different parts of the PCAT1-CtA complex.
**Figure supplement 4.** Resolution estimation, structure validation, and particle angular distribution.

**Table 2.** Summary of Cryo EM data and structure refinement statistics of PCAT1-CtA complex.

| Data collection | |
|---|---|
| Microscope | Titan Krios, 300 keV (FEI) |
| Detector | K2 summit direct electron detector (Gatan) |
| Energy Filter | 10 eV (Gatan) |
| Pixel size (Å) | 1.09 |
| number of movies | 3478 |
| Frames/movie | 60 |
| Total exposure time (s) | 12 |
| Exposure time per frame (s) | 0.2 |
| Total exposure (e/Å$^2$) | 80.8 |
| Defocus range (μM) | 0.8 to 2.2 |
| Final reconstruction | |
| Number of particles | 133698 |
| B-factor correction (Å$^2$) | −80 |
| RMS deviations | |
| bond length (Å) | 0.0028 |
| bond angles (°) | 1.2870 |
| Ramachandran | |
| favored (%) | 94.95 |
| Allowed (%) | 5.05 |
| Outlier (%) | 0.00 |
| Rotamer | |
| favored (%) | 87.30 |
| Allowed (%) | 9.73 |
| poor (%) | 2.97 |

(−7)XXXXG(−2)G(−1) (*Figure 3A*). Residues 8–25 of the leader peptide form a L-shaped structure that wraps around the PEP domain, with the N-terminus in the cytosol and the C-terminal end at the lateral opening of the translocation pathway (*Figure 3B and C*). Residues 15–21 (i.e, positions −10 to −4 of the consensus sequence) form a short helix, docking onto a hydrophobic groove on PEP (*Figures 3B* and *4A*). The double-glycine motif (G23 and G24), inserted to the active site of PEP, interacts with catalytic residues C21A and H99 (*Figure 3C*). The total buried surface of the leader peptide by the PEP domain is approximately 740 Å$^2$.

The substrate-binding cleft at the cleavage site is very narrow, which explains the specificity for the double-glycine motif in proteolysis. The Cβ atom of C21 (mutated to alanine) is about 5.5 Å away from G24, consistent with C21 serving as a nucleophile to attack the substrate backbone (*Figure 3C*, *Figure 3—figure supplement 1*) (*Schechter and Berger, 1967*). H99 orients the N atom of its imidazole ring towards the C21A, consistent with its role to polarize the attacking C21 (*Husain and Lowe, 1968*; *Wu and Tai, 2004*). The side chain of D115 forms a hydrogen bond with H99, maintaining H99 in an electronegative state and a catalytically favorable position (*Vernet et al., 1995*). In addition, the highly conserved Q15 residue is located in the vicinity, suggesting that it may function as an oxyanion hole that stabilizes the tetrahedral intermediate (*Ménard et al., 1991*). The configuration of the active site is typical of a cysteine protease, indicating that PCAT1 processes the CtA substrate via the common catalytic thiol mechanism (*Drenth et al., 1968*; *Kamphuis et al., 1984*).

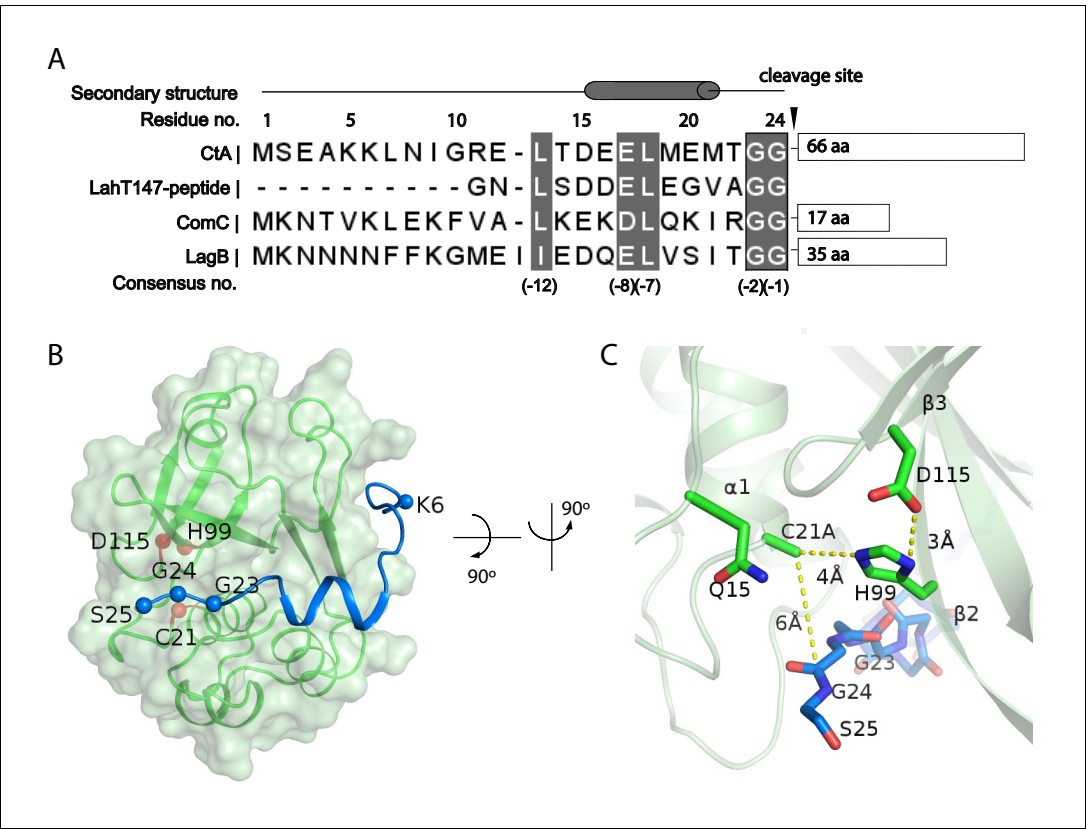

**Figure 3.** Substrate-PEP interaction. (**A**) Sequence alignment of different leader peptides. Consensus residues are highlighted in grey. (**B**) Docking of the leader peptide (blue) on the PEP surface (green). The PEP catalytic triad residues are indicated as red spheres and the CtA double glycine motif are indicated as blue spheres. (**C**) A zoomed-in view of the catalytic site.

The online version of this article includes the following figure supplement(s) for figure 3:

**Figure supplement 1.** Stereoview of the catalytic site.
**Figure supplement 2.** Comparison of PEP domains among PCATs.

## Substrate recognition

The structure of the PEP domain is highly conserved among different PCATs. The overall root-mean-square deviations (RMSDs) among the structures of ComA (*Ishii et al., 2010*) (PDB: 3K8U), LahT147 (*Bobeica et al., 2019*) (PDB: 6MPZ), and the PEP domain of PCAT1 are approximately 1 Å (*Figure 3—figure supplement 2*). In addition, substrates of the different PEPs share a consensus leader peptide sequence (*Figure 3A*), yet each PEP specifically recognizes one or a small subset of substrates.

What are the common substrate recognition motifs and what are the unique features that confer specificity? The leader peptides of CtA and the LahT147 substrate both contain a two-turn helix consisting of residues from positions −10 to −4, docking into a shallow groove on the surface of the PEP (*Figure 4A and B*). A hydrophobic knot, formed by the two leucine residues at positions −7 and −12 of the leader peptide and two hydrophobic residues on PEP, is observed in both structures (*Figure 4B*). To test the importance of this interaction, we mutated the conserved leucine residues in CtA and estimated its affinity for PCAT1(C21A) using a pull-down assay (*Figure 4D*). Compared to the *wt* CtA, the L(−7)A and L(−12)A mutants have much lower affinity for PCAT1. The pulled down PCAT1 could only be detected by Western blot even at the highest concentration of CtA tested (*Figure 4D*). We found hydrophobic residues A55 and I59 on PCAT1 that are within the Van der Waals radius of L(−7)A and L(−12)A, and are conserved in LahT147 as well (*Figure 4B*). Furthermore, reciprocal mutations introduced to residues A55 and I59 of PCAT1 lowered its affinity for CtA (*Figure 4E*) by more than eight fold, consistent with the role that these residues play on the CtA

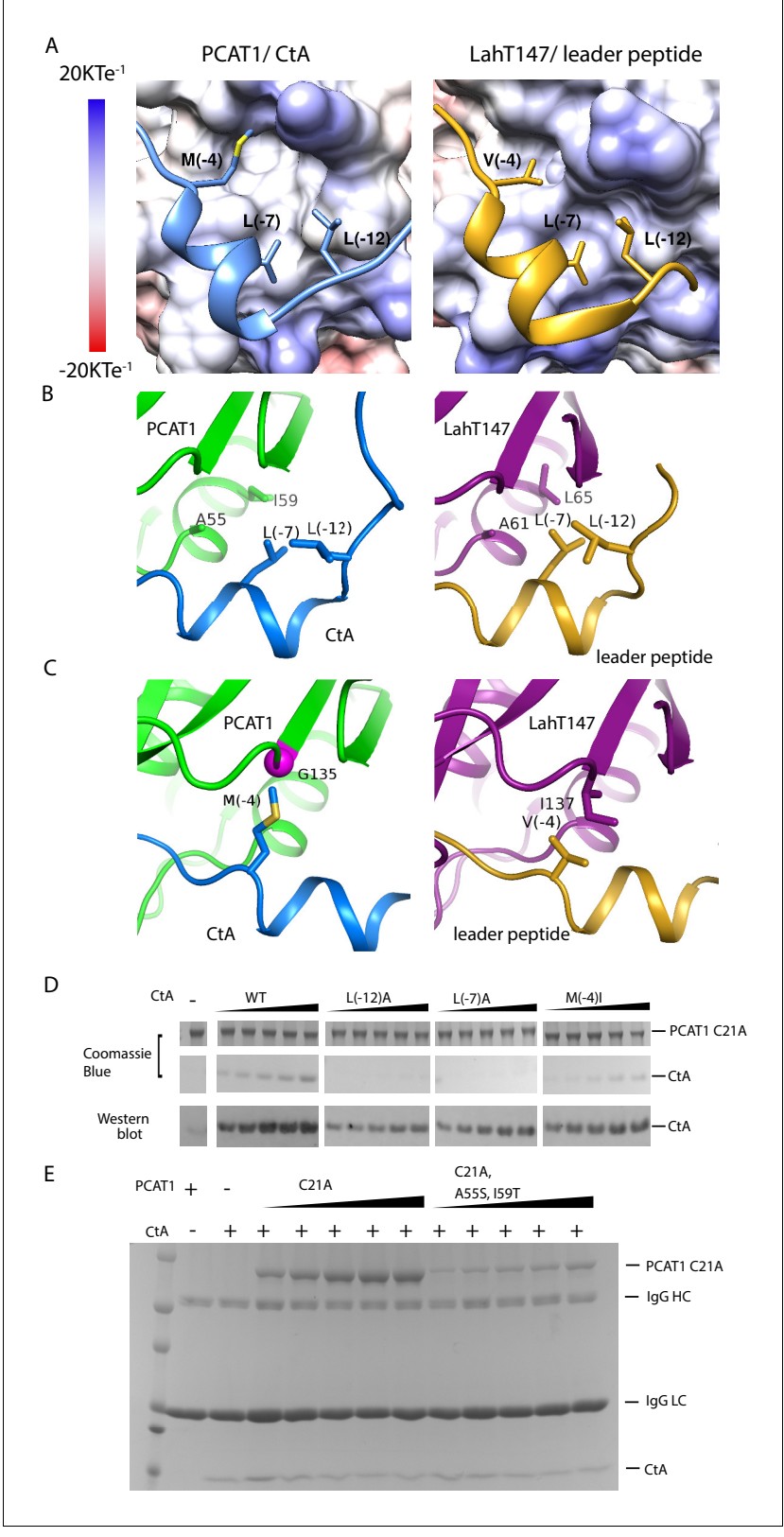

**Figure 4.** PCAT1 and LahT147 share common features in substrate recognition. (**A**) Insertion of three hydrophobic residues onto a hydrophobic groove on the surface of the PEP domain, represented as electrostatic potential surfaces. CtA is shown in blue and the LahT147 substrate in gold. (**B**) A conserved hydrophobic knot. (**C**) Residue at position −4 of the substrate conferring specificity. (**D**) Pulldown of the *wt* or mutant CtA using PCAT1(C21A)
*Figure 4 continued on next page*

*Figure 4 continued*

conjugated resin. The CtA constructs contain a 3x Flag tag at their C-terminus. Western blot was performed using the anti-Flag M2 antibody. (**E**) Pulldown of PCAT1(C21A) using the M2 antibody resin against the Flag-tagged CtA. Mutation of hydrophobic residue A55 and I59 were introduced to PCAT1(C21A) background. IgG HC and IgG LC denote the heavy chain and the light chain of the M2 antibody, respectively.

recognition. This hydrophobic interaction is likely a common feature shared by different PEP domains. Alanine substitution of either one of the two leucine residues in the ComA substrate decreased its affinity by approximately twenty fold (*Kotake et al., 2008*). Similarly, LahT147 does not process precursor peptides with substitutions at L(−7) or L(−12) (*Bobeica et al., 2019*).

Mutational data at the −4 position indicate that this residue is also important for the PEP-substrate interaction (*Figure 4D* and *Kotake et al., 2008*); however, in contrast to the highly conserved L(−7) and L(−12) residues, the amino acid identity at position −4 varies among different substrates (*Figure 3A and C*) and appears to correlate with the interacting residue on PEP. For example, M(−4) of CtA packs against G135 of PCAT1 (*Figure 4C*). In LahT147, a much smaller residue V(−4) of the substrate interacts with a larger residue I137 on PEP (*Figure 4C*). Therefore, it seems that the leader peptide position −4 confers specificity among different PCATs.

## Asymmetric positioning of the cargos

In contrast to the leader peptide, the cargo region (residues 25–90) is flexible and less defined. Inside the translocation pathway, we observe density of elongated shape packed along the interior surface of the cavity shown in the composite cryo-EM map of PCAT1 and CtA (*Figure 5A*, colored blue). To test if this density corresponds to the C-terminal region of CtA, single cysteines were introduced at six positions downstream of the leader peptide (*Figure 5B*). We also placed single cysteine, in three position, in the otherwise cysteine-free PCAT1: K275C and A433C on the interior surface of the TM cavity and K417C in an extracellular loop outside the translocation pathway (*Figure 5C*). The proximity of the cysteine pairs was analyzed by mixing CtA with PCAT1 in the presence of the oxidizing agent CuPhen or the reducing agent DTT, and then using Western blot to detect the appearance of PCAT1-CtA crosslinked product (*Figure 5B and C*). Mass spectrometry analysis was performed on three samples excised from the SDS-PAGE gel. In each case, the expected cross-linked peptide containing the engineered disulfide bond was observed. Cysteines at multiple positions along CtA can be crosslinked to K275C and A433C on PCAT1, suggesting that the cargo inside the cavity is unstructured and flexible, consistent with the amorphous nature of the EM density (*Figure 5A*). The cysteine at position 38 of CtA can be weakly crosslinked to K275C but not to K433C, which is consistent with the structure. K38 is near the leader peptide at the cytoplasmic side of the membrane and is not able to reach the extracellular ceiling of the cavity prior to proteolytic cleavage. No crosslinking product was observed for the cysteine placed outside the TM cavity (K417C), indicating that the reaction is specific (*Figure 5B and C*).

The density inside the TM cavity appears to be connected to only one of the two leader peptides (*Figure 5A and D*), raising the question whether one or two cargos are inside the TM cavity. We asked if the TM cavity is large enough to accommodate two cargos. Using the 3V server (*Voss and Gerstein, 2010*), the volume of the TM cavity is estimated to be 15,000 $Å^3$. Considering the average density of a small protein is around 1.35 g/cm (*Voss and Gerstein, 2010*; *Fischer et al., 2009*), the size of a CtA cargo is approximately 9000 $Å^3$, a conservative estimate for an unstructured protein. Thus, it is likely that only one cargo is inserted into the TM cavity. We termed the cargo inside the TM cavity the 'translocating CtA' as it is positioned for cleavage and translocation. The density for the cargo region is weak and amorphous, suggesting that there is no specific interaction between the cargo and transporter, and the substrate specificity is conferred through the leader peptide only. For the other substrate, the non-translocating CtA, no density is observed beyond S26, two residues beyond the double glycine motif (*Figure 5D* and *Figure 2—figure supplement 3E*). Furthermore, G23, G24, and S25 form a kink, orienting the C-terminus towards the cytoplasm (*Figure 5D* and *Figure 2—figure supplement 3E*). There is also biochemical evidence that the C-terminus of CtA is accessible: the PCAT1-CtA complex can be pulled down using an antibody against a tag placed at the C-terminus of CtA (*Lin et al., 2015*). Based on these observations, we

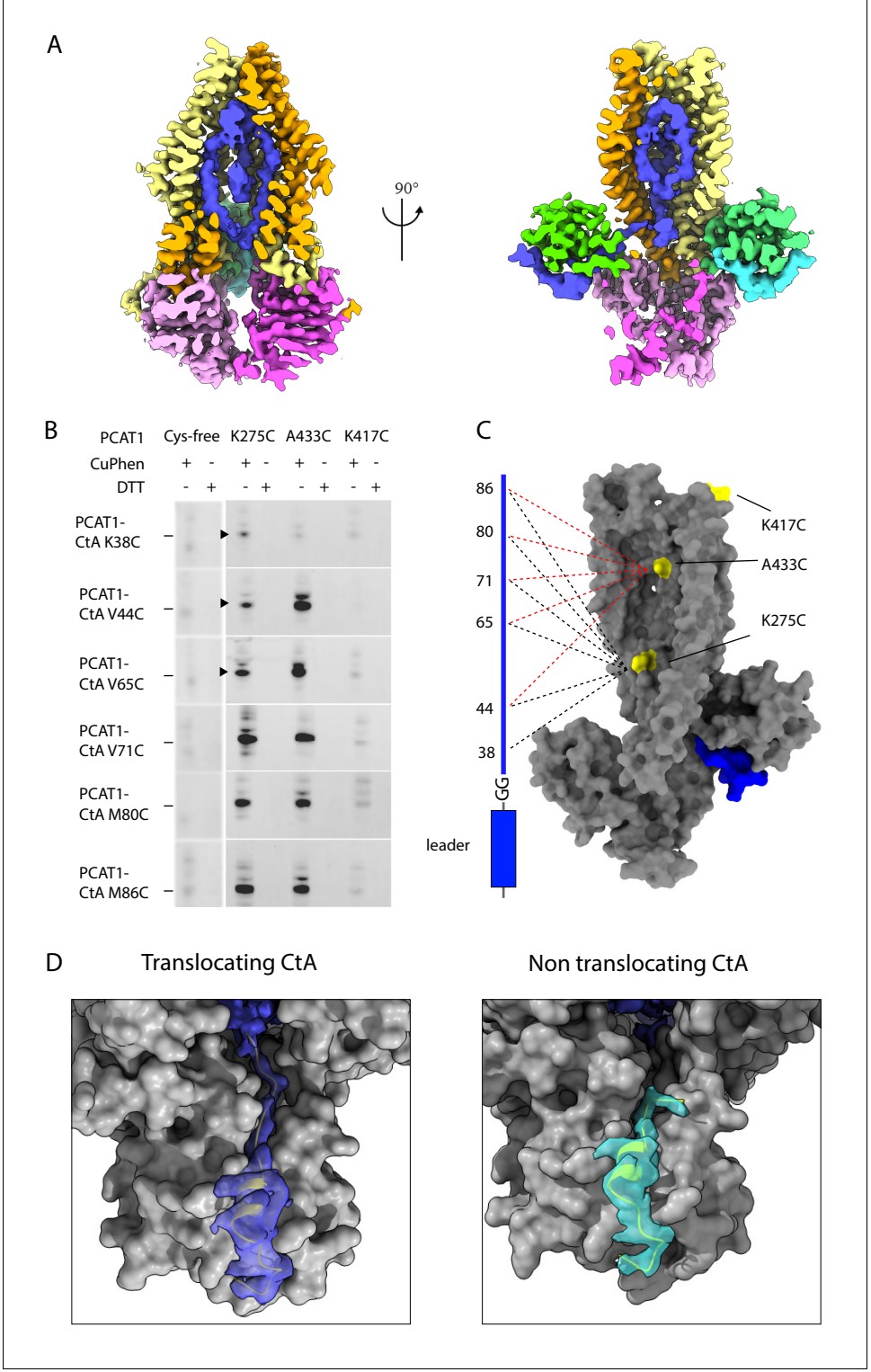

**Figure 5.** Positions of the two cargos. (**A**) Two orthogonal views of the the translocation pathway. A composite cryo-EM map displays PCAT1 density contoured at 0.5 using Chimera, the CtA contoured at 0.24, and the density inside the translocation pathway contoured at 0.15. The cryo-EM density inside the TM cavity (shown in blue) likely corresponds to the C-terminal region of one CtA molecule. (**B**) Disulfide crosslinking experiments between PCAT1 and CtA. Crosslinked PCAT1-CtA products were detected by Western blot using an anti-HA antibody against HA-tagged CtA. Three bands (indicated by arrows) were excised from the SDS-PAGE and analyzed by mass spectrometry. In all three cases, peptide fragments with the correct disulfide bond were identified. (**C**) Summary of

*Figure 5 continued on next page*

*Figure 5 continued*

the crosslinking results. A PCAT1 monomer is shown as grey surface and CtA is represented by a cartoon. The dotted lines illustrate the crosslinked pairs (black, 275C crosslink pairs; red, 433C crosslinked pairs). (**D**) Comparison of the cryo-EM density at the two catalytic sites. The density is displayed as surface. Dark blue, the translocating CtA; light blue, the non-translocating CtA. The maps were contoured at 0.6 using Chimera. The TMD and PEP are shown as grey surfaces and the NBDs are omitted for clarity. The leader peptides are shown as ribbons.

suggest that the non-translocating cargo is located in the cytosol, flexibly linked to the leader peptide.

## Conformational changes upon substrate binding

Comparing the structures of PCAT1 in the presence and absence of CtA shows that in the substrate-bound conformation, the intracellular opening of the TM cavity (formed from TM3 and TM4) is approximately 3 Å wider and the two NBDs are further apart (*Figure 6*). These structural differences are not influenced by crystal packing, as regions involved in lattice contacts do not undergo conformational changes upon CtA binding (*Figure 6—figure supplement 1*).

The conformational change of PCAT1 is opposite to that of the multidrug transporter MRP1, in which substrate binding brings the two NBDs closer (*Johnson and Chen, 2017*). MRP1, like many other ABC transporters, has higher ATPase activity in the presence of its substrate (*Johnson and Chen, 2017*). In contrast, addition of CtA reduces the ATPase activity of PCAT1 by about 10% for the wild-type protein and 50% for the cleavage-incompetent C21A mutant (*Lin et al., 2015*). As ATP hydrolysis requires NBD dimerization, the structural observation correlates well with the distinct biochemical property of this protein-conducting ABC transporter. The insertion of a large cargo into

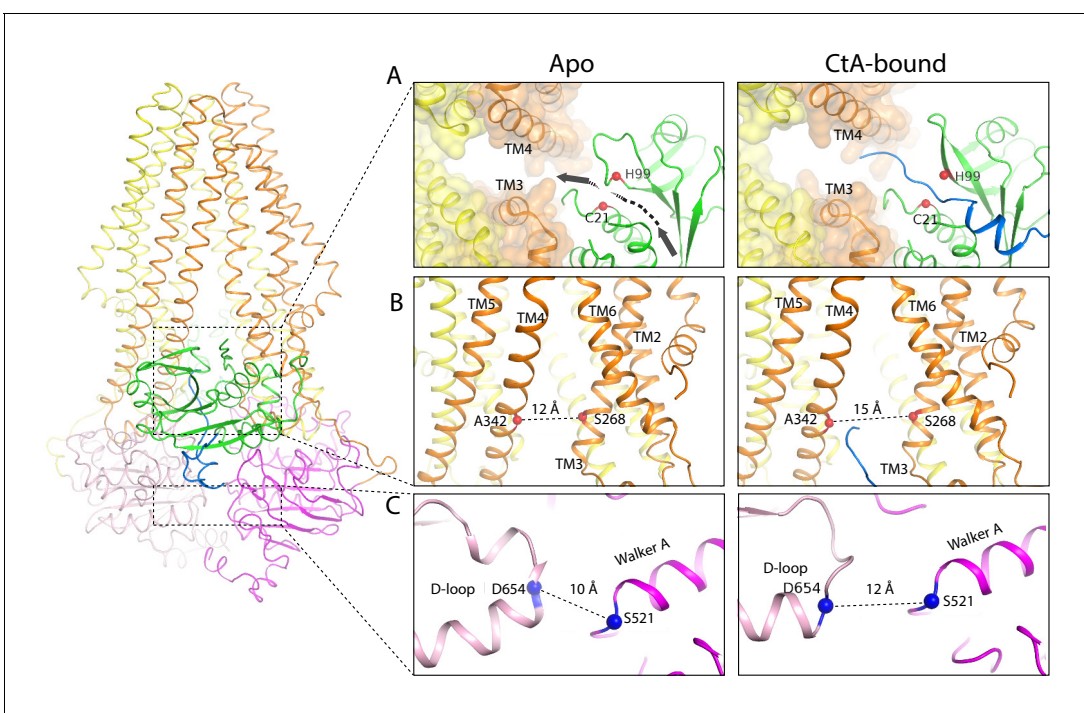

**Figure 6.** Conformational changes upon CtA binding. (**A,B**) Widening of the lateral opening of the TM pathway. The arrow in the apo structure delineates the CtA binding groove. The catalytic residues C21 and H99 are labeled. The Cα distances between residues S268 and A342 that line the lateral opening are shown. (**C**) The NBDs are further apart in the CtA-bound structure. The Cα distances between S521 in the walker A motif and D654 in the D-loop of the other NBD are indicated.

The online version of this article includes the following figure supplement(s) for figure 6:

**Figure supplement 1.** Crystal packing of *apo* PCAT1.

the TM cavity would likely decrease the rate of NBD dimerization thereby slowing down ATP hydrolysis.

Binding of the leader peptide also induces local conformational changes within the PEP domain. The largest displacement, approximately 5 Å, occurs in a loop (residues 91–98) preceding the catalytic residue H99 (*Figure 3—figure supplement 2C*). As a result, in the substrate-bound structure, a path between the catalytic site and the TM cavity opens up permitting insertion of the C-terminal cargo into the translocation pathway at the lateral opening between TM3 and TM4. (*Figure 6A*).

## Discussion

One of the key questions in the ABC transporter field is: How do transporters recruit and select substrates? The current understanding is that ABC importers largely depend on binding proteins to confer substrate selectivity while exporters interact with their substrate directly through their TMDs. The structures of ABC exporters, such as the multidrug transporters MRP1 (*Johnson and Chen, 2017*), ABCG2 (*Manolaridis et al., 2018*), P-glycoprotein (*Alam et al., 2019*), the Lipid A flipase MsbA (*Mi et al., 2017*), LPS transporter LptB$_2$FG (*Li et al., 2019*), and the promiscuous exporter TmrAB (*Hofmann et al., 2019*), all reveal a well-defined, substrate-binding site at the interface of the two TMDs. Here we present a different mechanism by which polypeptides are recognized by PCATs. Instead of the TMDs, the cytosolic peptidase domain binds substrate through its N-terminal secretion sequence, that is the leader peptide. The common feature shared by PCATs is a hydrophobic knot formed by two residues of the peptidase domain and residues at positions −7 and −12 of the leader peptide. Position −4 of the leader peptide likely confers substrate specificity for different PCATs.

Unlike most ABC exporters, the TMDs of PCAT1 do not provide any specific binding region for the cargo. Instead, they merely provide a large conduit to transverse the lipid bilayer. This observation explains why the PCAT system can be used to secrete different cargo proteins that are tethered to the same secretion signal (*van Belkum et al., 1997*). The absence of a high-affinity binding site in the TMDs is reminiscent of the *E. coli* cobalamin importer BtuCD, where the substrate is recruited by the periplasmic binding protein and the TMDs provide a teflon-like pathway for conduction (*Korkhov et al., 2012*; *Korkhov et al., 2014*).

The structure of the substrate-bound PCAT1 enables us to expand the previous working model of how polypeptides are processed and transported by PCATs in Gram-positive bacteria (*Figure 7*). PCAT1 adopts an inward-facing conformation in the absence of ATP, in which the NBD interface is open and each PEP domain docks onto a lateral opening of the TM pathway. Two substrates can bind to PCAT simultaneously, but only one substrate is positioned for cleavage and translocation. The translocating substrate inserts its C-terminal cargo into the TM cavity and the corresponding PEP domain cleaves the substrate to free the cargo. The other non-translocating cargo is located in

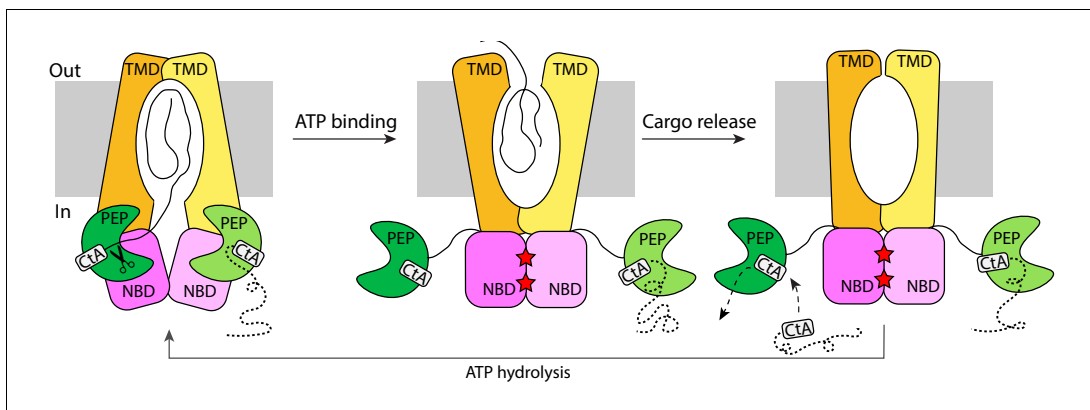

**Figure 7.** The alternating-access mechanism of PCAT1. CtA is recruited and cleaved in the inward-facing conformation. ATP binding stabilizes the outward-facing conformation in which the PEP domains are disengaged. After cargo release, TMDs isomerize to form an occluded cavity. ATP hydrolysis resets PCAT1 back to the inward-facing conformation, allowing PEP to dock into the TMD-NBD interface.

the cytosol. ATP binding brings the two NBDs into close contact, orienting the translocation pathway towards the extracellular space. As the TM pathway confers no specific binding of the substrate, the cargo will be released and, subsequently, the TMDs will isomerize to form the occluded conformation where both the extracellular gate and the cytoplasmic gate are closed (PDB: 4S0F). Formation of the closed NBD dimer disengages the PEP domains from the core transporter permitting release of the leader peptide and binding of a new substrate. ATP hydrolysis and ADP release reset the transporter to the inward-facing conformation allowing the PEP to dock back and position one of the two substrates for cleavage and translocation.

A key aspect of this model is the strict coupling of substrate cleavage to translocation. In the inward-facing conformation where two substrates are bound, the non-translocating substrate adopts a kink at the double-glycine motif, preventing uncoupled cleavage. In the ATP-bound conformation, the disengaged PEP domains have little proteolytic activity as the transporter core is necessary for substrate cleavage (*Lin et al., 2015*). This strict coupling is important to prevent the uncontrolled release of cargo molecules, some of which are toxic, into the cytosol of the secreting cells.

# Materials and methods

**Key resources table**

| Reagent type | Designation | Source | Identifier | Additional information |
|---|---|---|---|---|
| Strain (*Escherichia coli*) | BL21 (DE3)-RIL | Agilent | #230280 | |
| Chemical compound | n-Dodecyl-β-D-Maltopyranoside, Sol-Grade | Anatrace | D310S | |
| Chemical compound | n-Undecyl-β-D-Maltopyranoside, Anagrade | Anatrace | U300 | |
| Chemical compound | C8E4 | Anatrace | T350 | |
| Chemical compound | Phenanthroline | Sigma-Alrich | 131377 | |
| Commercial assay or kit | M2 agarose affinity gel | Sigma-Alrich | A4596 | |
| Commercial assay or kit | Superdex 200 increase | GE Lifesciences | 28990944 | |
| Commercial assay or kit | Glutathione Sepharose 4B resin | GE Healthcare | 17075604 | |
| Commercial assay or kit | anti-HA mouse monoclonal antibody | Invitrogen RRID:AB_10978021 | 26183 | 1:10000 |
| Commercial assay or kit | goat anti-mouse Alexa Fluor 680 | Invitrogen RRID:AB_2535723 | A21057 | 1:1000 |
| Commercial assay or kit | M2 anti-flag mouse antibody | Sigma-Alrich | F 1804–50 UG | 1:5000 |
| Other | R1.2–1.3 400 mesh Au holey carbon grids | Quantifoil | 1210627 | |
| Software, algorithm | Leginon | RRID:SCR_016731 | http://emg.nysbc.org/redmine/projects/leginon/wiki/Leginon_Homepage | |
| Software, algorithm | Unblur | DOI: 10.7554/eLife.06980 | http://grigorifflab.janelia.org/unblur | |
| Software, algorithm | CryoSPARC | DOI: 10.1038/nmeth.4169 RRID:SCR_016501 | https://cryosparc.com/ | |
| Software, algorithm | CTFFIND4 | DOI: 10.1016/j.jsb.2015.08.008 RRID:SCR_016732 | http://grigorifflab.janelia.org/ctf | |

*Continued on next page*

*Continued*

| Reagent type | Designation | Source | Identifier | Additional information |
|---|---|---|---|---|
| Software, algorithm | RELION | DOI: 10.7554/eLife.18722<br>DOI: 10.7554/eLife.41266<br>RRID:SCR_016274 | http://www2.mrc-lmb.cam.ac.uk/relion | |
| Software, algorithm | gCTF | DOI: 10.1016/j.jsb.2015.11.003<br>RRID:SCR_016500 | https://www.mrc-lmb.cam.ac.uk/kzhang/ | |
| Software, algorithm | Gautomatch | | https://www.mrc-lmb.cam.ac.uk/kzhang/ | |
| Software, algorithm | alignparts_lmbfgs | DOI: 10.1016/j.jsb.2015.08.007 | https://sites.google.com/site/rubinsteingroup/direct-detector-align_lmbfgs | |
| Software, algorithm | FrealignX | DOI: 10.1016/bs.mie.2016.04.013<br>RRID:SCR_016733 | http://grigoriefflab.janelia.org/frealign | |
| Software, algorithm | COOT | DOI: 10.1107/S0907444910007493<br>RRID:SCR_014222 | https://www2.mrc-lmb.cam.ac.uk/personal/pemsley/coot/ | |
| Software, algorithm | PHENIX | DOI: 10.1107/S0907444909052925<br>RRID:SCR_014224 | https://www.phenix-online.org | |
| Software, algorithm | MolProbity | DOI: 10.1107/S0907444909042073;<br>10.1093/nar/gkm216<br>RRID:SCR_014226 | http://molprobity.biochem.duke.edu | |
| Software, algorithm | Chimera | DOI: 10.1002/jcc.20084<br>RRID:SCR_004097 | https://www.cgl.ucsf.edu/chimera | |
| Software, algorithm | ChimeraX | DOI: 10.1002/pro.3235<br>RRID:SCR_015872 | https://www.https://www.rbvi.ucsf.edu/chimerax/ | |
| Software, algorithm | Pymol | PyMOL<br>RRID:SCR_000305 | http://www.pymol.org | |
| Software, algorithm | APBS | DOI: 10.1093/nar/gkm276<br>RRID:SCR_008387 | http://www.poissonboltzmann.org | |
| Software, algorithm | 3V web server | DOI: 10.1093/nar/gkq395 | http://3vee.molmovdb.org/ | |
| Software, algorithm | UniDec v 3.2 | DOI: 10.1021/acs.analchem.5b00140 | https://github.com/michaelmarty/UniDec | |

## Protein expression and purification

The wt and mutant *PCAT1* genes were subcloned into the pMCSG20 vector with an N-terminal glutathione-S-transferase (GST) tag and a Tobacco Etch Virus (TEV) protease cleavage site. Protein was expressed and purified as previously described (*Lin et al., 2015*). Briefly, *E. coli* strain BL21(DE3) codon plus (RIL) cells expressing PCAT1 were lysed and solubilized in buffer containing 1% n-dodecyl-β-D-maltoside (DDM; Anatrace), 50 mM Tris pH 7.0, 500 mM NaCl, 10% glycerol, and 5 mM DTT. PCAT1 was enriched on Glutathione Sepharose 4B resin, washed with buffer containing 50 mM Tris pH 7.0, 500 mM NaCl, 10% glycerol, 5 mM DTT, and 2 mM n-undecyl-β-D-maltopyranoside (UDM; Anatrace). The GST tag was removed by cleavage with TEV protease, and PCAT1 was further purified using a Superdex 200 increase column (GE Healthcare) in a buffer containing 50 mM Tris pH 7.0, 150 mM NaCl, 2 mM UDM.

The gene encoding CtA was subcloned into the pMCSG7 vector with an N-terminal TEV-cleavable 6x His tag and a C-terminal 3x Flag or HA-tag. RIL cells expressing CtA were resuspended in lysis buffer (50 mM Tris pH 7.0, 150 mM NaCl, and 10% glycerol), lysed by three passes through a high-pressure homogenizer (Emulsiflex C-3; Avestin), and centrifuged at 80,000 g for 40 min to isolate inclusion bodies. The pellet was washed extensively with lysis buffer plus 1% Triton X-100, then with lysis buffer alone, before solubilizing in 8 M urea. The denatured protein was purified on cobalt affinity resin (Clontech Laboratories) and refolded via dialysis in lysis buffer plus 5 mM DTT. The His-tag was removed by cleavage with TEV protease and the protein were further purified by gel-filtration chromatography (Superdex 75 HiLoad 16/60, GE Healthcare).

## Native mass spectrometry analysis

The purified PCAT1 samples were buffer exchanged into native MS solution (200 mM ammonium acetate pH 7.5 with either 0.058% UDM or 0.5% $C_8E_4$) using Zeba microspin desalting columns (Thermo Scientific) with a 40 kDa molecular weight cut-off (MWCO). The typical concentrations used for native MS analysis were 4 µM PCAT1 monomer for the PCAT1-only sample and 4 µM PCAT1 (C21A) monomer + 8 µM CtA (a two-fold excess of CtA). An aliquot (2–3 µL) of the sample was loaded into a gold-coated quartz capillary that was fabricated in-house. The sample was then electrosprayed into an Exactive Plus EMR instrument (Thermo Fisher Scientific) using a static nanospray source. The MS parameters used include: spray voltage, 1.0–1.4 kV; capillary temperature, 125°C; S-lens RF level, 200; resolving power, 8750 or 17,500 at m/z of 200; AGC target, $1 \times 10^6$; number of microscans, 5; maximum injection time, 200 ms; injection flatapole, 8 V; interflatapole, 4 V; bent flatapole, 4 V; ultrahigh vacuum pressure, $7–10 \times 10^{-10}$ mbar; total number of scans, 100. The insource dissociation (ISD) and higher-energy collisional dissociation (HCD) parameters were varied and optimized accordingly (see *Figure 1—figure supplements 1* and *2*). Mass calibration was performed using cesium iodide. The acquired MS spectra were visualized using Thermo Xcalibur Qual Browser (version 3.0.63) and deconvolution was performed either manually or using UniDec version 3.2 (*Marty et al., 2015*; *Reid et al., 2019*). The deconvolved spectra from UniDec were plotted using the m/z software (Proteometrics LLC). Experimental masses were reported as the average mass ± standard deviation (S.D.) across all calculated mass values obtained within the observed charge state series.

## Cryo-EM sample preparation and data collection

Purified PCAT1(C21A) (5 mg/ml, 62 µM monomer) was mixed with 62 µM refolded refolded CtA and incubated on ice for 30 mins. About 3 µl of sample was applied onto glow-discharged holey carbon grids (Quantifoil gold R1.2–1.3), incubated for 20 s at 100% humidity, and blotted with filter paper for 3 s before being plunge-frozen into liquid ethane using a Vitrobot Mark IV (FEI). A dataset of 3478 movies was collected on the Titan Krios Transmission Electron Microscope (FEI) outfitted with a K2 Summit direct electron detector (Gatan) with a super-resolution pixel size of 0.545 Å using Leginon (*Suloway et al., 2009*). The electron dose rate was eight electrons/pixel/s with a total exposure time of 12 s resulting in a total electron dose of 80 electrons/Å$^2$ over 60 frames.

## Image processing

The procedure for image processing is summarized in *Figure 2—figure supplement 1* . Movie frames were corrected using a gain reference and binned by a factor of two to a pixel size of 1.09 Å. Movie frames were aligned using Unblur (*Grant and Grigorieff, 2015*) and the effective contrast transfer function (CTF) was determined from frame-summed micrographs using CTFFIND4 (*Rohou and Grigorieff, 2015*). Templates for auto-picking were generated from 2D classes generated from 5000 manually picked particles in RELION (*Scheres, 2016*). After manual inspection to remove false positives, 572,800 automated picked particles were extracted with a box size of 300 pixels and subjected to drift correction using alignparts_lmbfgs (*Rubinstein and Brubaker, 2015*). The resulting particles were 2D-classified into 150 classes after which 383,002 particles were selected. An *ab initio* 3D model with C2 symmetry, generated from CryoSPARC (*Punjani et al., 2017*), was low-passed filtered to 60 Å and used as an initial model for 3D Classification in RELION (*Scheres, 2016*). The most populated class was further refined in RELION to 4 Å resolution. A smoothed mask, excluding the detergent micelle, was created and used for 3D classification without alignment and subsequent local refinement in RELION. The final 3D reconstruction with C2 symmetry yielded a 3.9 Å map.

The movie frames were also motion-corrected using MotionCor2 software (*Zheng et al., 2017*), and CTF estimation was calculated using gCTF both implemented in Relion 3 (*Zhang, 2016*). Subsequently, the final set of selected particle were re-extracted and subjected to three iterations of Bayesian particle polishing (*Zivanov et al., 2019*), CTF refinement (*Zivanov et al., 2018*), and local refinement with C2 symmetry in Relion 3. One round of refinement in C1 was performed to release the symmetry from C2 to C1. Subsequent 3D classification without image alignment was performed. The 3D classes were manually inspected for differences in local asymmetry. The particles belonging to a 3D class that appeared to have an opposite orientation were rotated 180° along the symmetry

axis. One round of local refinement with C1 symmetry yielded 3.6 Å resolution in Relion. The final round of refinement was performed using CryoSPARC produces a map of 3.37 Å resolution (*Punjani et al., 2017*). The resolution was estimated using Fourier Shell Correlation (FSC) with 0.143 cutoff implemented in the 3DFSC web application (*Tan et al., 2017*) *Figure 2—figure supplement 4*. Local resolution estimation from two CryoSPARC half maps was performed in CryoSPARC *Figure 2—figure supplement 2*.

## Model building and refinement

The crystal structure of PCAT1 (PDB:4RY2) was placed into a sharpened cryo-EM map (sharpening factor, $-80$ Å$^2$) using rigid body fitting in Chimera (*Pettersen et al., 2004*) followed by manual adjustments in Coot (*Emsley and Cowtan, 2004*). The final model consists of residues 9–722 of PCAT1, residues 8–29 of the translocating CtA, and residues 8–25 of the non-translocating CtA.

The model was initially refined against one working half map in real-space by PHENIX (*Adams et al., 2010*), followed by rounds of refinement in reciprocal space using REFMAC (*Brown et al., 2015*), with secondary structure and reference restraints derived from ProSMART (*Nicholls et al., 2014*). The quality of the final model was evaluated by MolProbity (*Chen et al., 2010*). To assess the degree of overfitting, we calculated the FSC curves between the model and working half map, the non-working half map, and the full map using SPIDER (*Frank et al., 1996*).

## Disulfide cross-linking accessibility assay

To perform disulfide cross-linking accessibility assay, we constructed a cysteine-free PCAT1 by replacing all nine cysteines in PCAT1 with serine. Single cysteine substitutions at positions 275, 417, or 433 were introduced to the cysteine-free PCAT1 construct by site-directed mutagenesis. Mutant CtA (0.6 µM) containing an introduced cysteine residue and a HA tag was mixed with equimolar single-cysteine PCAT1 in buffer containing 50 mM HEPES pH 7.0, 150 mM NaCl, and 2 mM UDM. The reaction mixture was incubated at room temperature for 10 min in the presence of 15 µM Cu-Phenanthroline or 50 mM DTT before being analyzed by SDS-PAGE. For Western blotting, the cross-linked product was visualized with a primary anti-HA mouse monoclonal antibody (1:10000) and a secondary goat anti-mouse Alexa Fluor 680 antibody (1:1000). In addition, we have verified the identity of the cross-linked products by mass spectrometry for the following pairs: PCAT1 K275C-CtA K38C, PCAT1 K275C-CtA K44C, and PCAT1 K275C-CtA K65C. MS analysis was performed by the Rockefeller University Proteomics Resource Center.

## Pull-down assays

To assess residues on CtA for PEP binding, PCAT1 (C21A)-bound GST sepharose resin was used to pull down wt or mutant CtA. CtA titration was performed by mixing 20 µl of Glutathione resin with 0.2, 0.4, 0.8, 1.6, or 3.2 nmol of CtA in 100 µl reaction buffer. The reaction mixture was incubated for 30 min on ice, after which the resin was washed twice with 400 µl wash buffer (50 mM Tris pH 7.0, 500 mM NaCl, 10% glycerol, 2 mM UDM, 5 mM DTT). The samples were analyzed by SDS-PAGE as well as Western with M2 anti-flag mouse antibody as primary antibody (1:5000) and goat anti-mouse Alexa Fluor 680 antibody as a secondary antibody (1:1000).

To assess the reciprocal binding residues on PCAT1, site-directed mutagenesis was performed to introduce mutation to PCAT1 (C21A), after which the pulldown experiment was performed using the C-terminal 3xFlag tagged CtA as a bait. For each reaction, 0.6 µM of PCAT1 was incubated on ice for 10 min with 1.2, 2.4, 4.8, 9.6, or 19.2 µM of CtA in 100 µl reaction buffer (50 mM HEPES pH 7.0, 150 mM NaCl, 2 mM UDM, 5 mM DTT). To capture the PCAT1 (C21A)-CtA complex, 15 µl of Anti-Flag M2 agarose affinity gel (Sigma-Aldrich) was added to the reaction and incubated for 30 min. The agarose affinity gel was washed twice with 400 µl reaction buffer. Samples analyzed by SDS-PAGE.

## Figure preparation

Structure figures were prepared using the program PyMOL (*Schrodinger, 2015*), UCSF Chimera (*Pettersen et al., 2004*), and UCSF ChimeraX (*Goddard et al., 2018*).

## Acknowledgements

We thank Mark Ebrahim and Johanna Sotiris at the Rockefeller Evelyn Gruss Lipper Cryo-Electron Microscopy Resource Center and Edward Eng at the Simons Electron Microscopy Center, The New York Structural Biology Center, for assistance in data collection. This work was supported by the Howard Hughes Medical Institute (to JC) and the National Institutes of Health grants P41 GM109824 and P41 GM103314 (to BTC). The authors declare no competing financial interests.

## Additional information

### Funding

| Funder | Grant reference number | Author |
|---|---|---|
| Howard Hughes Medical Institute | | Jue Chen |
| National Institutes of Health | P41 GM109824 | Brian T Chait |
| National Institutes of Health | P41 GM103314 | Brian T Chait |

The funders had no role in study design, data collection and interpretation, or the decision to submit the work for publication.

### Author contributions

Virapat Kieuvongngam, Formal analysis, Validation, Investigation, Visualization, Project administration; Paul Dominic B Olinares, Resources, Formal analysis, Validation, Investigation, Visualization; Anthony Palillo, Resources; Michael L Oldham, Resources, Data curation, Validation; Brian T Chait, Conceptualization, Data curation, Supervision, Funding acquisition; Jue Chen, Conceptualization, Data curation, Supervision, Funding acquisition, Project administration

### Author ORCIDs

Virapat Kieuvongngam https://orcid.org/0000-0002-9614-0619
Paul Dominic B Olinares https://orcid.org/0000-0002-3429-6618
Jue Chen https://orcid.org/0000-0003-2075-4283

### Decision letter and Author response

Decision letter https://doi.org/10.7554/eLife.51492.sa1
Author response https://doi.org/10.7554/eLife.51492.sa2

## Additional files

### Supplementary files

• Transparent reporting form

### Data availability

Cryo-EM density map of PCAT1-CtA complex has been deposited into electron microscopy data bank (EMDB) under accession code EMD-21132. Atomic coordinates of PCAT1-CtA complex has been deposited in the protein data bank (PDB) under accession code 6V9Z.

The following datasets were generated:

| Author(s) | Year | Dataset title | Dataset URL | Database and Identifier |
|---|---|---|---|---|
| Kieuvongngam V, Oldham ML, Chen J | 2019 | Atomic coordinates of PCAT1-CtA complex | https://www.rcsb.org/structure/6V9Z | RCSB Protein Data Bank, 6V9Z |
| Virapat Kieuvongngam, Michael L Oldham, Jue Chen | 2019 | Cryo-EM structure of PCAT1 bound to its CtA peptide substrate | https://www.ebi.ac.uk/pdbe/entry/emdb/EMD-21132 | EMDB, EMD-21132 |

The following previously published datasets were used:

| Author(s) | Year | Dataset title | Dataset URL | Database and Identifier |
|---|---|---|---|---|
| Lin DL, Huang S, Chen J | 2014 | Crystal structure of the peptidase-containing ABC transporter PCAT1 | https://www.rcsb.org/structure/4RY2 | RCSB Protein Data Bank, 4RY2 |
| Lin DL, Huang S, Chen J | 2014 | Crystal structure of the peptidase-containing ABC transporter PCAT1 E648Q mutant complexed with ATPgS in an occluded conformation | https://www.rcsb.org/structure/4S0F | RCSB Protein Data Bank, 4S0F |
| Ishii S, Yano T, Ebihara A, Okamo-to A, Manzoku M, Hayashi H | 2009 | Crystal Structure of the Peptidase Domain of Streptococcus ComA, a Bi-functional ABC Transporter Involved in Quorum Sensing Pathway | https://www.rcsb.org/structure/3K8U | RCSB Protein Data Bank, 3K8U |
| Dong S-H, Nair SK | 2018 | Crystal structure of a double glycine motif protease from AMS/PCAT transporter in complex with the leader peptide | https://www.rcsb.org/structure/6MPZ | RCSB Protein Data Bank, 6MPZ |

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
