## [Decision Letter]

**Acceptance summary:**

In Gram-positive bacteria, a class of ABC transporters known an PCATs transport antimicrobial or quorum-sensing peptides; these peptides are produced by the transporter itself through proteolytic cleavage of a precursor polypeptide, by means of peptidase (PEP) domains fused to the transporter transmembrane segment. The study by Chen and co-workers reveals the cryo-EM structure of PCAT1 in complex with its substrate CtA. This work builds upon the author's previously determined crystal structure of apo PCAT1. The new structure, together with native mass spectrometry and cross-linking data, demonstrate that PCAT1 can recognize up to two substrate molecules via both of its peptidase domains, yet only one of these peptides is translocated in each transport cycle. The study constitutes a significant advancement not only in regards our understanding of the PCAT system; more broadly, the work provides insights into a mechanism whereby two distinct activities within a given membrane system become coupled – namely proteolysis and active transport.

**Decision letter after peer review:**

Thank you for submitting your article "Structural basis of substrate recognition by a polypeptide processing and secretion transporter" for consideration by *eLife*. Your article has been reviewed by three peer reviewers, and the evaluation has been overseen by José D Faraldo-Gómez as Reviewing Editor and Olga Boudker as the Senior Editor. The following reviewers have agreed to reveal their identity: Jochen Zimmer (Reviewer #1); and Show-Ling Shyng (Reviewer #2).

The reviewers have discussed the reviews with one another and the Reviewing Editor has drafted this decision to help you prepare a revised submission.

Summary:

The manuscript by Kieuvongngam et al. presents the cryo-EM structure of the PCAT1 ABC transporter in complex with its substrate CtA. In Gram-positives, PCATs secrete antimicrobial or quorum-sensing peptides, which are produced by the transporter upon proteolytic cleavage of a precursor polypeptide. Accordingly, PCAT1 contains peptidase (PEP) domains fused to the transporter transmembrane segment. The work builds upon the author's previously determined crystal structure of apo PCAT1. The new structure, together with native mass spectrometry and cross-linking data, demonstrate that PCAT1 can recognize up to two substrate molecules via both of its peptidase domains, yet only one of these peptides is translocated in each transport cycle. Reviewers found that the study was "well designed and executed" and that the findings presented "will be of great interest" to others working in the molecular membrane physiology, particularly on ABC transporters. The reviewers also pointed that the study is "well written and the data nicely presented". This positive feedback notwithstanding, reviewers and editors agree that several important revisions are essential for the manuscript to be acceptable for publication in *eLife*.

Essential revisions:

1) Cryo-EM workflow

The authors state that 2D (and 3D) classifications were used to remove false positive and low-resolution particles and that a subset of 102699 particles were selected in the last refinements. Yet, the authors also report that they used the entire set of autopicked particles (572800) for the final 3D reconstruction. It is a critical concern that this set is highly likely to include a significant number of false positives, which end up contributing to Einstein-noise. The authors should also note that by using Frealign in the final refinement step, the resolution is not calculated according to the gold-standard, as it is in Relion. Instead, in Frealign or CisTEM all particles are used during refinement and only later separated into two halves for FSC calculation. This procedure makes it difficult to assess the influence of noise in the final resolution calculation – which makes our concern in regard to the particle selection the more relevant. The authors are therefore asked to redo the final reconstruction with a more selective set of particles, and to explain their rationale. In addition, if the Frealign/CisTEM is used for the final refinement, the authors ought to clarify that the resolution was not calculated according to the gold-standard, and include additional details, e.g. the Mw and the resolution of the reference used during refinement.

2) Disulfide cross-linking of CtA and PCAT1

This experiment should include several controls demonstrating that the observed high-molecular weight species are indeed CtA-PCAT1 cross-links. As such, a Cys-less PCAT1 in combination with the various CtA constructs should not yield detectable species and neither should Cys-less CtA with the selected PCAT1 constructs. In addition, all products should be sensitive to DTT reduction. Some of the cross-linked species are also detected with the K417C control, suggesting that some oxidation occurs after SDS-denaturation. Blocking with NEM prior to SDS denaturation might eliminate this background. Samples could be probed with both anti-FLAG and anti-PCAT1 to make it clear the band(s) are crosslinked CtA-PCAT1.

---

## [Author Response]

Essential revisions:1) Cryo-EM workflowThe authors state that 2D (and 3D) classifications were used to remove false positive and low-resolution particles and that a subset of 102699 particles were selected in the last refinements. Yet, the authors also report that they used the entire set of autopicked particles (572800) for the final 3D reconstruction. It is a critical concern that this set is highly likely to include a significant number of false positives, which end up contributing to Einstein-noise. The authors should also note that by using Frealign in the final refinement step, the resolution is not calculated according to the gold-standard, as it is in Relion. Instead, in Frealign or CisTEM all particles are used during refinement and only later separated into two halves for FSC calculation. This procedure makes it difficult to assess the influence of noise in the final resolution calculation – which makes our concern in regard to the particle selection the more relevant. The authors are therefore asked to redo the final reconstruction with a more selective set of particles, and to explain their rationale. In addition, if the Frealign/CisTEM is used for the final refinement, the authors ought to clarify that the resolution was not calculated according to the gold-standard, and include additional details, e.g. the Mw and the resolution of the reference used during refinement.

We thank the reviewers for this suggestion. We have re-processed the data as requested and obtained a better-quality map using a subset of particles instead of the full dataset. Based on the “the gold-standard” implemented in CryoSPARC, the final reconstruction has an estimated resolution of 3.4 Å, significantly better than the previous reconstruction (3.7 Å). The revised cryo-EM analysis procedure is described in the Materials and methods section and in Figure 2—figure supplement 1-4.

2) Disulfide cross-linking of CtA and PCAT1This experiment should include several controls demonstrating that the observed high-molecular weight species are indeed CtA-PCAT1 cross-links. As such, a Cys-less PCAT1 in combination with the various CtA constructs should not yield detectable species and neither should Cys-less CtA with the selected PCAT1 constructs. In addition, all products should be sensitive to DTT reduction. Some of the cross-linked species are also detected with the K417C control, suggesting that some oxidation occurs after SDS-denaturation. Blocking with NEM prior to SDS denaturation might eliminate this background. Samples could be probed with both anti-FLAG and anti-PCAT1 to make it clear the band(s) are crosslinked CtA-PCAT1.

Controls, including Cys-less PCAT1 and reducing condition (+ DTT), are now included in Figure 5B. In addition, we have verified the identity of the cross-linked product by mass spectrometry for the following pairs: PCAT1 K275C-CtA K38C, PCAT1 K275C-CtA K44C, and PCAT1 K275C-CtA K65C. These results are described in the revision:

“Mass spectrometry analysis was performed on three samples excised from the SDS-PAGE. In all cases, peptide fragments cross-linked through the engineered disulfide bond were detected.”